# Online Normalization for Training Neural Networks

**Vitaliy Chiley**[*]    **Ilya Sharapov**[*]    **Atli Kosson**    **Urs Koster**

**Ryan Reece**    **Sofía Samaniego de la Fuente**    **Vishal Subbiah**    **Michael James**[* †]

Cerebras Systems
175 S. San Antonio Road
Los Altos, California 94022

## Abstract

Online Normalization is a new technique for normalizing the hidden activations of a neural network. Like Batch Normalization, it normalizes the sample dimension. While Online Normalization does not use batches, it is as accurate as Batch Normalization. We resolve a theoretical limitation of Batch Normalization by introducing an unbiased technique for computing the gradient of normalized activations. Online Normalization works with automatic differentiation by adding statistical normalization as a primitive. This technique can be used in cases not covered by some other normalizers, such as recurrent networks, fully connected networks, and networks with activation memory requirements prohibitive for batching. We show its applications to image classification, image segmentation, and language modeling. We present formal proofs and experimental results on ImageNet, CIFAR, and PTB datasets.

## 1    Introduction

Traditionally, neural networks are *functions* that map inputs deterministically to outputs. Normalization makes this non-deterministic because each sample is affected not only by the network weights but also by the statistical distribution of samples. Therefore, normalization re-defines neural networks to be *statistical operators*. Normalized networks treat each neuron's output as a random variable that ultimately depends on the network's parameters and input distribution. No matter how it is stimulated, a normalized neuron produces an output distribution with zero mean and unit variance.

While normalization has enjoyed widespread success, current normalization methods have theoretical and practical limitations. These limitations stem from an inability to compute the gradient of the ideal normalization operator.

Batch methods are commonly used to approximate ideal normalization. These methods use the distribution of the current minibatch as a proxy for the distribution of the entire dataset. They produce biased estimates of the gradient that violate a fundamental tenet of stochastic gradient descent (SGD): It is not possible to recover the true gradient from any number of small batch evaluations. This bias becomes more pronounced as batch size is reduced.

Increasing the minibatch size provides more accurate approximations of normalization and its gradient at the cost of increased memory consumption. This is especially problematic for image processing and volumetric networks. Here neural activations outnumber network parameters, and even modest batch sizes reduce the trainable network size by an order of magnitude.

---

[*]Equal contribution
[†]Corresponding author: `michael@cerebras.net`

Online Normalization is a new algorithm that resolves these limitations while matching or exceeding the performance of current methods. It computes unbiased activations and unbiased gradients without any use of batching. Online Normalization differentiates through the normalization operator in a way that has theoretical justification. We show the technique working at scale with the ImageNet [1] ResNet-50 [2] classification benchmark, as well as with smaller networks for image classification, image segmentation, and recurrent language modeling.

Instead of using batches, Online Normalization uses running estimates of activation statistics in the forward pass with a corrective guard to prevent exponential behavior. The backward pass implements a control process to ensure that back-propagated gradients stay within a bounded distance of true gradients. A geometrical analysis of normalization reveals necessary and sufficient conditions that characterize the gradient of the normalization operator. We further analyze the effect of approximation errors in the forward and backward passes on network dynamics. Based on our findings we present the Online Normalization technique and experiments that compare it with other normalization methods. Formal proofs and all details necessary to reproduce results are in the appendix. Additionally we provide reference code in PyTorch, TensorFlow, and C [3].

## 2   Related work

Ioffe and Szegedy introduced normalization of hidden activations [4], defining it as a transformation that uses full dataset statistics to eliminate *internal covariate shift*. They observed that the inability to differentiate through a running estimator of forward statistics produces a gradient that leads to divergence [5]. They resolved this with the Batch Normalization method [4]. During training, each minibatch is used as a statistical proxy for the entire dataset. This allows use of gradient descent without a running estimator process. However, training still maintains running estimates for use during validation and inference.

The success of Batch Normalization has inspired a number of related methods that address its limitations. They can be classified as functional or heuristic methods.

Functional methods replace the normalization operator with a normalization function. The function is chosen to share certain properties of the normalization operator. Layer Normalization [6] normalizes across features instead of across samples. Group Normalization [7] generalizes this by partitioning features into groups. Weight Normalization [8] and Normalization Propagation [9] apply normalization to network weights instead of network activations.

The advantage of functional normalizers is that they fit within the SGD framework, and work in recurrent networks and large networks. However, when compared directly to batch normalization they generally perform worse [7].

Heuristic methods use measurements from previous network iterations to augment the current forward and backward passes. These methods do not differentiate through the normalization operator. Instead, they combine terms from previous batch-based approximations. An advantage of heuristic normalizers is that they use more data to generate better estimates of forward statistics; however, they lack correctness and stability guarantees.

Batch Renormalization [5] is one example of a heuristic method. While it uses an online process to estimate dataset statistics, these estimates are based on batches and are only allowed to be within a fixed interval of the current batch's statistics. Batch Renormalization does not differentiate through its statistical estimation process, and like Instance Normalization [10], it cannot be used with fully connected layers at a batch size of one.

Streaming Normalization [11] is also a heuristic method. It performs one weight update for every several minibatches. Instead of differentiating through the normalization operator, it averages point gradients at long and short time scales. It applies a different mixture in a saw-tooth pattern to each minibatch depending on its timing relative to the latest weight update.

In recurrent networks, circular dependencies between sample statistics and activations pose a challenge to normalization [12, 13, 14]. Recurrent Batch Normalization [12] offers the approach of maintaining distinct statistics for each time step. At inference this results in a different linear operation being applied at each time step, breaking the formalism of recurrent networks. Functional normalizers avoid circular dependencies and have been shown to perform better [6].

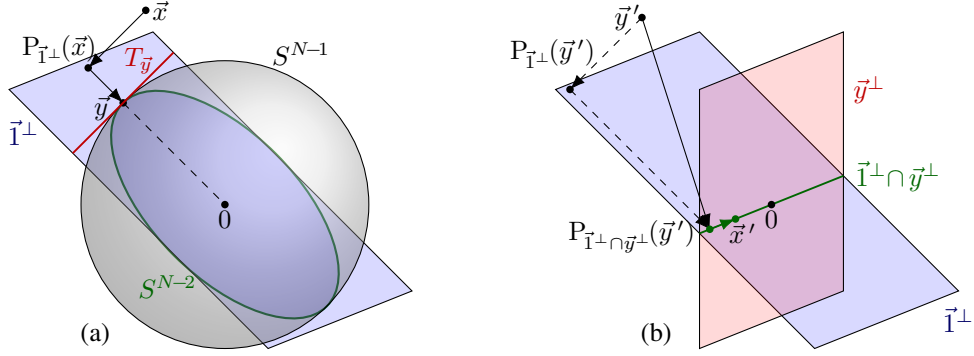

Figure 1: Geometry of normalization.

## 3 Principles of normalization

Normalization is an affine transformation $f_{\mathbb{X}}$ that maps a scalar random variable $x$ to an output $y$ with zero mean and unit variance. It maps every sample in a way that depends on the distribution $\mathbb{X}$,

$$f_{\mathbb{X}}[x] \equiv \frac{x - \mu[x]}{\sigma[x]} \qquad x \sim \mathbb{X}\,, \tag{1}$$

resulting in normalized output $y$ satisfying

$$\mu[y] = 0 \quad \text{and} \quad \mu[y^2] = 1\,. \tag{2}$$

When we apply normalization to network activations, the input distribution $\mathbb{X}$ is itself functionally dependent on the state of the network, in particular on the weights of all prior layers. This poses a challenge for accurate computation of normalization because at no point in time can we observe the entire distribution corresponding to the current values of the weights.

Backpropagation uses the chain rule to compute the derivative of the loss function $L$ with respect to hidden activations. We express this using the convention $(\cdot)' = \partial L / \partial (\cdot)$ as

$$x' = \frac{\partial f_{\mathbb{X}}[x]}{\partial x}[y']\,. \tag{3}$$

It is not obvious how to handle the derivative in the preceding equation, which is itself a statistical operator. The usual approaches do not work: Automatic differentiation cannot be applied to expectations. Exact computation over the entire dataset is prohibitive. Ignoring the derivative causes a feedback loop between gradient descent and the estimator process, leading to instability [4].

Batch Normalization avoids these challenges by freezing the network while it measures the statistics of a batch. Increasing batch size improves accuracy of the gradients but also increases memory requirements and potentially impedes learning. We started our study with the question: Is freezing the network the only way to resolve interference between an estimator process and gradient descent? It is not. In the following sections we will show how to achieve the asymptotic accuracy of large batch normalization while inspecting only one sample at a time.

### 3.1 Properties of normalized activations and gradients

Differential geometry provides key insights on normalization. Let $\vec{x} \in \mathbb{R}^N$ be a finite-dimensional vector whose components approximate the normalizer's input distribution. In the geometric setting, normalization is a *function* defined on $\mathbb{R}^N$. Its output $\vec{y}$ satisfies both conditions of (2). The zero mean condition is satisfied on the subspace $\vec{1}^{\perp}$ orthogonal to the ones vector, whereas the unit variance condition is satisfied on the sphere $S^{N-1}$ with radius $\sqrt{N}$ (Figure 1a). Therefore $\vec{y}$ lies on the manifold $S^{N-2} = \vec{1}^{\perp} \cap S^{N-1}$.

Clearly, mapping $\mathbb{R}^N$ to a sphere is nonlinear. The forward pass (1) does this in two steps: It subtracts the same value from all components of $\vec{x}$, which is orthogonal projection $P_{\vec{1}^{\perp}}$; then it rescales the

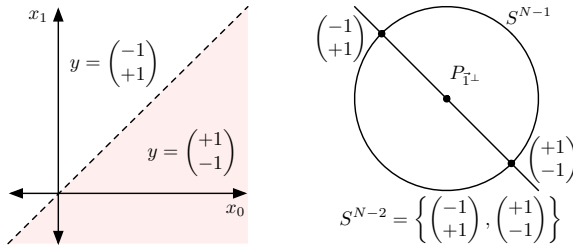
Figure 2: Two element normalization (N=2).

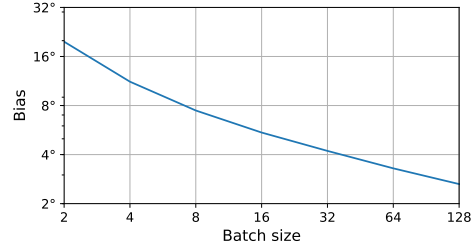
Figure 3: Gradient bias (BN).

result to $S^{N-1}$. In contrast, the backward pass (3) is linear because the chain rule produces a product of Jacobians. The Jacobian $\mathrm{J} = [\partial y_j/\partial x_i]$ must suppress gradient components that would move $\vec{y}$ off the manifold's tangent space. $S^{N-2}$ is a sphere embedded in a subspace, so its tangent space $T_{\vec{y}}$ at $\vec{y}$ is orthogonal to both the sphere's radius $\vec{y}$ and the subspace's complement $\vec{1}$.

$$\vec{x}\,' = \mathrm{J}\vec{y}\,' \implies \mathrm{P}_{\vec{1}}(\vec{x}\,') = \mathrm{P}_{\vec{y}}(\vec{x}\,') = 0 \; . \tag{4}$$

Because (1) is the composition of two steps, J is a product of two factors (Figure 1b). The unbiasing step $\mathrm{P}_{\vec{1}\perp}$ is linear and therefore is also its own Jacobian. The scaling step is isotropic in $\vec{y}^\perp$ and therefore its Jacobian acts equally to all components in $\vec{y}^\perp$ scaling them by $\sigma$. The remaining $\vec{y}$ component must be suppressed (4), resulting in:

$$\mathrm{J} = \frac{1}{\sigma}\,\mathrm{P}_{\vec{1}\perp}\mathrm{P}_{\vec{y}\perp} \implies \vec{x}\,' = \frac{1}{\sigma}\left(\mathrm{I} - \mathrm{P}_{\vec{1}}\right)\left(\mathrm{I} - \mathrm{P}_{\vec{y}}\right)\vec{y}\,' \; . \tag{5}$$

This is the exact expression for backpropagation through the normalization operator. It is also possible to reach the same conclusion algebraically [5] (Appendix B).

The input $\vec{x}$ is a continuous function of the neural network's weights and dataset distribution. During training, the incremental weight updates cause $\vec{x}$ to drift. Meanwhile, normalization is only presented with a single scalar component of $\vec{x}$ while the other components remain unknown. Online Normalization handles this with an online control process that examines a single sample per step while ensuring (5) is always approximately satisfied throughout training.

## 3.2 Bias in gradient estimates

Although normalization applies an affine transformation, it has a nonlinear dependence on the input distribution $\mathbb{X}$. Therefore, sampling the gradient of a normalized network with mini-batches results in biased estimates. This effect becomes more pronounced for smaller mini-batch sizes. Consider the extreme case of normalizing a fully connected layer with batch size two (Figure 2). Each pair of samples is transformed to either $(-1, +1)$ or $(+1, -1)$, resulting in a piecewise constant surface. Since the output is discrete, the corresponding gradient is zero almost everywhere. Of course, the true gradient is nonzero almost everywhere and therefore cannot be recovered from any number of batch-two evaluations.

The same effect can be seen in more realistic cases. Figure 3 shows gradient bias as a function of batch size measured for a convolutional network with the CIFAR-10 dataset [15]. Ground truth for this plot used all 50,000 images in the dataset with weights randomly initialized and fixed. Even in this simple scenario, moderate batch sizes exhibit bias exceeding an angle of 10 degrees.

## 3.3 Exploding and vanishing activations

All normalizers are presented with the task of calculating specific values of the affine coefficients $\mu[x]$ and $\sigma[x]$ for the forward pass (1). Exact computation of these coefficients is impossible without processing the entire dataset. Therefore, SGD-based optimizers must admit errors in normalization statistics. These errors are problematic for networks that have unbounded activation functions, such as ReLU. It is possible for the errors to amplify through the depth of the network causing exponential growth of activation magnitudes.

Figure 4 shows exponential behavior for a 100-layer fully connected network with a synthetic dataset. In each layer we compute exact affine coefficients using the entire dataset. We randomly perturb

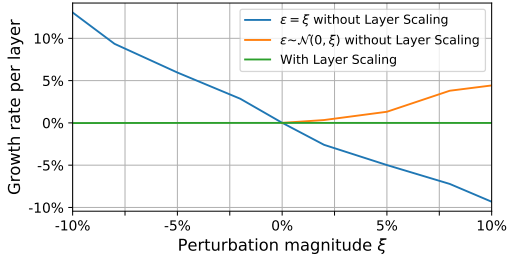
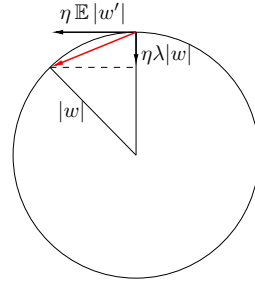

Figure 4: Activation growth.　　　　　　　Figure 5: Weight equilibrium.

the coefficients before applying inference to assess the sensitivity to errors. Exponential behavior is easy to observe even with mild noise. This effect is particularly pronounced when variances $\sigma^2$ are systematically underestimated, in which case each layer amplifies the signal in expectation.

Batch Normalization does not exhibit exponential behavior. Although its estimates contain error, exact normalization of a batch of inputs imposes (2) as strict constraints on normalized output. For each layer, the largest possible output component is bounded by the square root of the batch size. Exponential behavior is precluded because this bound does not depend on the depth of the network. This property is also enjoyed by Layer Normalization and Group Normalization.

Any successful online procedure will also need a mechanism to avoid exponential growth of activations. With a bounded activation function, such as tanh, this is achieved automatically. *Layer scaling* (Figure 4) that enforces the second equality of (2) across all features in a layer is another possible mechanism that prevents both growth and decay of activations.

### 3.4 Invariance to gradient scale

When a normalizer follows a linear layer, the normalized output is invariant to the scale of the weights $|w|$ [5, 6]. Scaling the weights by any constant is immediately absorbed by the normalizer. Therefore, $\partial y / \partial |w|$ is zero and gradient descent makes steps orthogonal to the weight vector (Figure 5). With a fixed learning rate $\eta$, a sequence of steps of size $O(\eta)$ leads to unbounded growth of $|w|$. Each successive step will have decreasing relative effect on the weight change reducing the effective learning rate.

Others have observed that the $L_2$ weight decay [16] commonly used in normalized networks counteracts the growth of $|w|$. In particular, [17] analyzes this phenomenon, although under a faulty assumption that gradients are not backpropagated through the mean and variance calculations. Instead, we observe that weight growth and decay are balanced when weights reach an equilibrium scale (Figure 5). We denote the gradient with respect to weights $w'$ and the increment in weights $\Delta w \equiv \eta w'$. When $\eta$ and decay factor $\lambda$ are small, solving for equilibrium yields (Appendix C):

$$|w| = \sqrt{\frac{\eta}{2\lambda}} \, \mathbb{E}\,|w'| \; . \tag{6}$$

The equilibrium weight magnitude depends on $\eta$. When the weights are away from their equilibrium magnitude, such as at initialization and after each learning rate drop, the weights tend to either grow or diminish network-wide. This tendency can create a biased error in statistical estimates that can lead to exponential behavior (Section 3.3).

Scale invariance with respect to the weights means that the learning trajectory depends only on the ratio $\Delta w / |w|$ and the problem can be arbitrarily reparametrized as long as this ratio is kept constant. This shows that $L_2$ weight decay does not have a regularizing effect; it only corrects for the radial growth artifact introduced by the finite step size of SGD.

When weights are in the equilibrium described by (6),

$$\frac{\Delta w}{|w|} = \sqrt{2\eta\lambda} \, \frac{w'}{\mathbb{E}\,|w'|} \; . \tag{7}$$

This equation shows that learning dynamics are invariant to the scale of the distribution of gradients $\mathbb{E}\,|w'|$. We also observe that the effective learning rate is $\sqrt{2\eta\lambda}$. This correspondence was indepen-

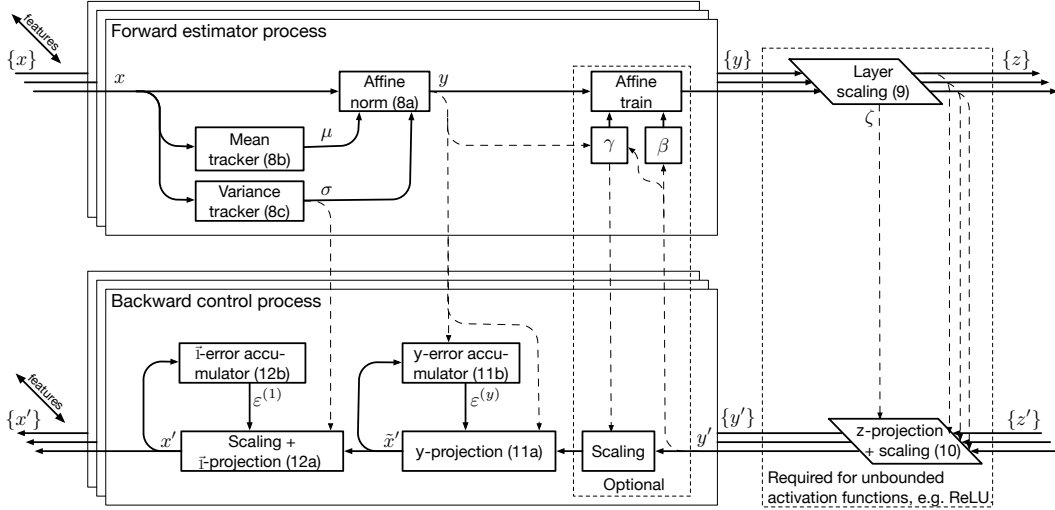

Figure 6: Online Normalization.

dently observed by Page [18]. Practitioners tend to use linear scaling of the learning rate with batch size [19] while keeping the $L_2$ regularization constant $\lambda$ fixed. Equation (7) shows that this amounts to the square root scaling suggested earlier by Krizhevsky [20].

## 4 Online Normalization

To define Online Normalization (Figure 6), we replace arithmetic averages over the full dataset in (2) with exponentially decaying averages of online samples. Similarly, projections in (4) and (5) are computed over online data using exponentially decaying inner products. The decay factors $\alpha_{\mathrm{f}}$ and $\alpha_{\mathrm{b}}$ for forward and backward passes respectively are hyperparameters for the technique.

We allow incoming samples $x_t$, such as images, to have multiple scalar components and denote feature-wide mean and variance by $\mu(x_t)$ and $\sigma^2(x_t)$. The algorithm also applies to outputs of fully connected layers with only one scalar output per feature. In fact, this case simplifies to $\mu(x_t) = x_t$ and $\sigma(x_t) = 0$. We use scalars $\mu_t$ and $\sigma_t$ to denote running estimates of mean and variance across all samples. The subscript $t$ denotes time steps corresponding to processing new incoming samples.

Online Normalization uses an ongoing process during the forward pass to estimate activation means and variances. It implements the standard online computation of mean and variance [21, 22] generalized to processing multi-value samples and exponential averaging of sample statistics. The resulting estimates directly lead to an affine normalization transform.

$$y_t = \frac{x_t - \mu_{t-1}}{\sigma_{t-1}} \tag{8a}$$

$$\mu_t = \alpha_{\mathrm{f}}\mu_{t-1} + (1 - \alpha_{\mathrm{f}})\mu(x_t) \tag{8b}$$

$$\sigma_t^2 = \alpha_{\mathrm{f}}\sigma_{t-1}^2 + (1 - \alpha_{\mathrm{f}})\sigma^2(x_t) + \alpha_{\mathrm{f}}(1 - \alpha_{\mathrm{f}})\left(\mu(x_t) - \mu_{t-1}\right)^2 \tag{8c}$$

This process removes two degrees of freedom for each feature that may be restored adding another affine transform with adaptive bias and gain. Corresponding equations are standard in normalization literature [4] and are not reproduced here. The forward pass concludes with a layer-scaling stage that uses data from all features to prevent exponential growth (Section 3.3):

$$z_t = \frac{y_t}{\zeta_t} \quad \text{with} \quad \zeta_t = \sqrt{\mu(\{y_t^2\})}\,, \tag{9}$$

where $\{\cdot\}$ includes all features.

The backward pass proceeds in reverse order, starting with the exact gradient of layer scaling:

$$y_t' = \frac{z_t' - z_t\mu(\{z_t z_t'\})}{\zeta_t}\,. \tag{10}$$

| Table 1: Memory for training (GB). | | | |
|---|---|---|---|
| Network | Online Norm | Batch 32 | 128 |
| ResNet-50, ImageNet | 1 | 2 | 4 |
| ResNet-50, PyTorch[a] | 2 | 5 | 15 |
| U-Net, $150^3$ voxels | 1 | 29 | 115 |
| U-Net, $250^3$ voxels | 6 | 195 | 785 |
| U-Net, $1024^2$ pixels | 2 | 31 | 123 |
| U-Net, $2048^2$ pixels | 5 | 137 | 546 |

[a] PyTorch stores multiple copies of activations for improved performance.

| Table 2: Best validation: loss (accuracy%). | | | |
|---|---|---|---|
| Normalizer | CIFAR-10 ResNet-20 | CIFAR-100 ResNet-20 | ImageNet ResNet-50 |
| Online | **0.26** (**92.3**) | **1.12** (**68.6**) | **0.94** (76.3) |
| Batch[a] | **0.26** (92.2) | 1.14 (**68.6**) | 0.97 (**76.4**) |
| Group | 0.32 (90.3) | 1.35 (63.3) | (75.9)[b] |
| Instance | 0.31 (90.4) | 1.32 (63.1) | (71.6)[b] |
| Layer | 0.39 (87.4) | 1.47 (59.2) | (74.7)[b] |
| Weight | - | - | (67 )[b] |
| Propagation | - | - | (71.9)[b] |

[a] Batch size 128 for CIFAR and 32 for ImageNet.
[b] Data from [7, 23, 24].

The backward pass continues through per-feature normalization (8) using a control mechanism to back out projections defined by (5). We do it in two steps, controlling for orthogonality to $\vec{y}$ first

$$\tilde{x}'_t = y'_t - (1 - \alpha_b)\varepsilon^{(y)}_{t-1} y_t \tag{11a}$$

$$\varepsilon^{(y)}_t = \varepsilon^{(y)}_{t-1} + \mu(\tilde{x}'_t y_t) \tag{11b}$$

and then for the mean-zero condition

$$x'_t = \frac{\tilde{x}'_t}{\sigma_{t-1}} - (1 - \alpha_b)\varepsilon^{(1)}_{t-1} \tag{12a}$$

$$\varepsilon^{(1)}_t = \varepsilon^{(1)}_{t-1} + \mu(x'_t) \; . \tag{12b}$$

Gradient scale invariance (Section 3.4) shows that scaling with the running estimate of input variance $\sigma_t$ in (12a) is optional and can be replaced by rescaling the output $x'_t$ with a running average to force it to the unit norm in expectation.

**Formal Properties** Online Normalization provides arbitrarily good approximations of ideal normalization and its gradient. The quality of approximation is controlled by the hyperparameters $\alpha_f$, $\alpha_b$, and the learning rate $\eta$. Parameters $\alpha_f$ and $\alpha_b$ determine the extent of temporal averaging and $\eta$ controls the rate of change of the input distribution. Online Normalization also satisfies the gradient's orthogonality requirements. In the course of training, the accumulated errors $\varepsilon^{(y)}_t$ and $\varepsilon^{(1)}_t$ that track deviation from orthogonality (5) remain bounded. Formal derivations are in Appendix D.

**Memory Requirements** Networks that use Batch Normalization tend to train poorly with small batches. Larger batches are required for accurate estimates of parameter gradients, but activation memory usage increases linearly with batch size. This limits the size of models that can be trained on a given system. Online Normalization achieves same accuracy without requiring batches (Section 5). Table 1 shows that using batches for classification of 2D images leads to a considerable increase in the memory footprint; for 3D volumes, batching becomes prohibitive even with modestly sized images.

## 5 Experiments

We demonstrate Online Normalization in a variety of settings. In our experience it has ported easily to new networks and tasks. Details for replicating experiments as well as statistical characterization of experiment reproducibility are in Appendix A. Scripts to reproduce our results are in the companion repository [3].

*CIFAR image classification (Figures 7-8, Table 2).* Our experiments start with the best-published hyperparameter settings for ResNet-20 [2] for use with Batch Normalization on a single GPU. We accept these hyperparameters as fixed values for use with Online Normalization. Online Normalization introduces two hyperparameters, decay rates $\alpha_f$ and $\alpha_b$. We used a logarithmic grid sweep to determine good settings. Then we ran five independent trials for each normalizer. Online Normalization had the best validation performance of all compared methods.

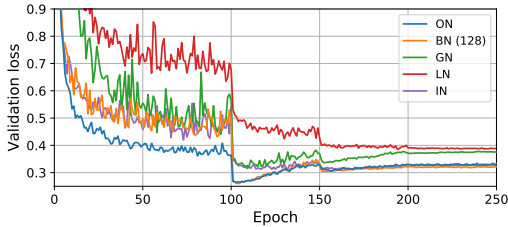

Figure 7: CIFAR-10 / ResNet-20.

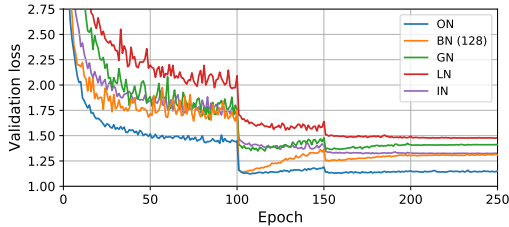

Figure 8: CIFAR-100 / ResNet-20.

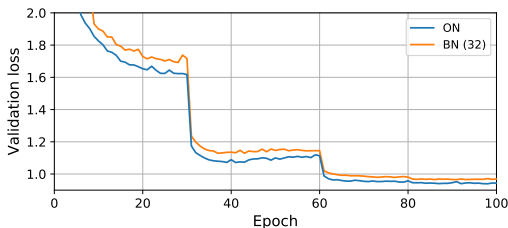

Figure 9: ImageNet / ResNet-50.

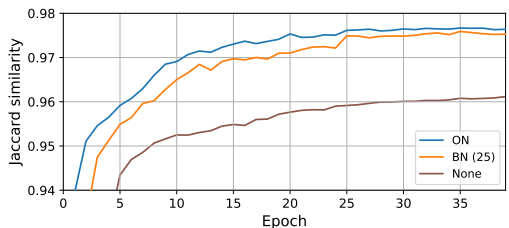

Figure 10: Image Segmentation with U-Net.

*ImageNet image classification (Figure 9, Table 2).* For the ResNet-50 [2] experiment, we are reporting the single experimental run that we conducted. This trial used decay factors chosen based on the CIFAR experiments. Even better results should be possible with a sweep. Our training procedure is based on a protocol tuned for Batch Normalization [25]. Even without tuning, Online Normalization achieves the best validation loss of all methods. At validation time it is nearly as accurate as Batch Normalization and both methods are better than other compared methods.

*U-Net image segmentation (Figure 10).* The U-Net [26] architecture has applications in segmenting 2D and 3D images. It has been applied to volumetric segmentation in 3D scans [27]. Volumetric convolutions require large memories for activations (Table 1), making Batch Normalization impractical. Our small-scale experiment performs image segmentation on a synthetic shape dataset [28]. Online Normalization achieves the best Jaccard similarity coefficient among compared methods.

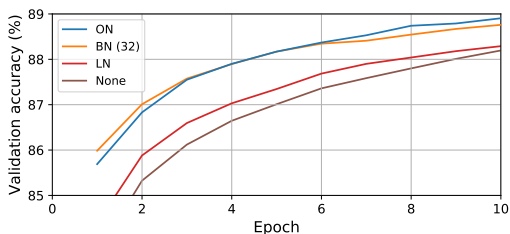

Figure 11: FMNIST with MLP.

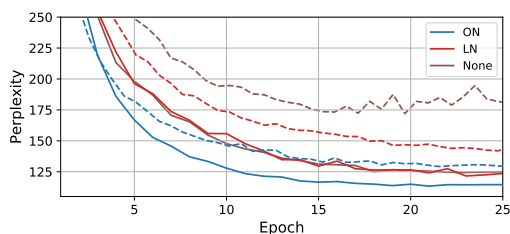

Figure 12: RNN (dashed) and LSTM (solid).

*Fully-connected network (Figure 11).* Online Normalization also works when normalizer inputs are single scalars. We used a three-layer fully connected network, 500+300 HU [29], for the Fashion MNIST [30] classification task. Fashion MNIST is a harder task than MNIST digit recognition, and therefore provides more discrimination power in our comparison. The initial learning trajectory shows Online Normalization outperforms the other normalizers.

*Recurrent language modeling (Figure 12).* Online Normalization works without modification in recurrent networks. It maintains statistics using information from all previous samples and time steps. This information is representative of the distribution of all recurrent activations, allowing Online Normalization to work in the presence of circular dependencies (Section 2). We train word based language models of PTB [31] using single layer RNN and LSTM. The LSTM network uses normalization on the four gate activation functions, but not the memory cell. This allows the memory cell to encode a persistent state for unbounded time without normalization forcing it to zero mean. In both the RNN and LSTM, Online Normalization performs better than the other methods. Remarkably, the RNN using Online Normalization performs nearly as well as the unnormalized LSTM.

# 6 Conclusion

Online Normalization is a robust normalizer that performs competitively with the best normalizers for large-scale networks and works for cases where other normalizers do not apply. The technique is formally derived and straightforward to implement. The gradient of normalization is remarkably simple: it is only a linear projection and scaling.

There have been concerns in the field that normalization violates the paradigm of SGD [5, 8, 9]. A main tenet of SGD is that noisy measurements can be averaged to the true value of the gradient. Batch normalization has a fundamental gradient bias dependent on the batch size that cannot be eliminated by additional averaging or reduction in the learning rate. Because Batch Normalization requires batches, it leaves the value of the gradient for any individual input undefined. This within-batch computation has been seen as biologically implausible [11].

In contrast, we have shown that the normalization operator and its gradient can be implemented locally within individual neurons. The computation does not require keeping track of specific prior activations. Additionally, normalization allows neurons to locally maintain input weights at any scale of choice–without coordinating with other neurons. Finally any gradient signal generated by the neuron is also scale-free and independent of gradient scale employed by other neurons. In aggregate ideal normalization (1) provides stability and localized computation for all three phases of gradient descent: forward propagation, backward propagation, and weight update. Other methods do not have this property. For instance, Layer Normalization requires layer-wide communication and Batch Normalization is implemented by computing within-batch dependencies.

We expect normalization to remain important as the community continues to explore larger and deeper networks. Memory will become even more precious in this scenario. Online Normalization enables batch-free training resulting in over an order of magnitude reduction of activation memory.

### Acknowledgments

We thank Rob Schreiber, Gary Lauterbach, Natalia Vassilieva, Andy Hock, Scott James and Xin Wang for their help and comments that greatly improved the manuscript. We thank Devansh Arpit for insightful discussions. We also thank Natalia Vassilieva for modeling memory requirements for U-Net and Michael Kural for work on this project during his internship.

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
