[Supplementary Material · online_normalization_appendix.pdf]

# Appendix A  Experimental details

We give an overview of experimental details for the results presented in the paper. All experiments were performed on Amazon's EC2 P3 single GPU instances.

## A.1  ResNet

We train ResNet using the SGD with momentum optimizer. $L_2$ regularization is applied. A learning rate decay factor is applied at predefined epochs. Training procedure and hyperparameters are adapted from [25].

For CIFAR10 and CIFAR100 training, we adopt the hyperparameters optimized for training using Batch Normalization. Performing a hyperparameter search for the network with Online Normalization is expected to produce better results. We perform a logarithmic sweep from $1/2$ through $4095/4096$ to set the forward and backward decay factors $\alpha_f$ and $\alpha_b$. Then we perform five independent runs for the network with Batch Normalization and Online Normalization. The results shown in Figure 7-8 are a median of the five independent results.

We conduct and report only a single experimental run for ImageNet training. When using Batch Normalization, the optimal hyperparameters for training ImageNet are given in [2] where training was done at batch size 256. We train our network using batch sizes appropriate for single GPU training. The momentum and learning rate hyperparameters are adapted using the scaling rules found in Appendix F. For training ResNet with Online Normalization we use the same hyperparameters used for training with Batch Normalization and set decay factors based on CIFAR10 experiments. Performing a hyperparameter search for all hyperparameters is expected to produce better performance.

All hyperparameters are summarized in Table 3.

Table 3: ResNet Training Hyperparameters.

| Dataset<br>Network | ImageNet<br>ResNet50 | CIFAR10<br>ResNet20 | CIFAR100<br>ResNet20 |
|---|---|---|---|
| Epochs | 100 | 250 | 250 |
| Batch size | 32 | 128 | 128 |
| Learning rate ($\eta$) | 0.01308 | 0.1 | 0.1 |
| Optimizer momentum ($\mu$) | 0.98692 | 0.9 | 0.9 |
| $L_2$ constant ($\lambda$) | $10^{-4}$ | $2 \times 10^{-4}$ | $2 \times 10^{-4}$ |
| LR decay factor | 0.1 | 0.1 | 0.1 |
| LR decay epochs | {30, 60, 80, 90} | {100, 150, 200} | {100, 150, 200} |
| Forward decay factor ($\alpha_f$) | .999 | $1023/1024$ | $511/512$ |
| Backward decay factor ($\alpha_b$) | .99 | $127/128$ | $15/16$ |

## A.2  U-Net

U-Net is trained updating parameters at an update cadence of 25. Training is done for 40 epochs using the SGD with momentum optimizer on a synthetic image dataset [28]. L2 regularization is applied. A learning rate (LR) decay factor is applied at epoch 25. The dataset uses 2000 samples in the training set and 200 samples in the validation set. Synthetic dataset generation and model definition are adapted from [28]. U-Net is trained using no normalization, Batch Normalization and Online Normalization. Normalization is added before each ReLU as in [27]. Learning rate, $\eta = m \times 10^{-n}$, sweeps are performed on the network with no normalization and on the network with Batch Normalization. $m$ and $n$ are swept in the ranges 0 to 9 and 0 to 5 respectively using a step size of 1. We use Online Normalization as a drop-in replacement for Batch Normalization. The network with Online Normalization uses the learning rate found to perform optimally in the network with Batch Normalization. Logarithmic sweeps from $15/16$ to $32767/32768$ and $1/2$ to $8191/8192$ are performed to set the forward and backward decay factors respectively. All hyperparameters are summarized in Table 4.

For U-Net training, and subsequent examples, we observe relatively high run to run variability because the datasets are small. Training the network without normalization produced a few outliers which show poor average performance. We report the median of 50 runs (Figure 10); reporting the mean would unfairly misrepresent the network without normalization as having poor expected performance.

Table 4: U-Net Training Hyperparameters.

| Normalizer | ON | BN | - |
|---|---|---|---|
| Learning rate ($\eta$) | 0.04 | 0.04 | 0.6 |
| Optimizer momentum ($\mu$) | 0.9 | 0.9 | 0.9 |
| $L_2$ constant ($\lambda$) | $10^{-6}$ | $10^{-6}$ | $10^{-6}$ |
| LR decay factor | 0.1 | 0.1 | 0.1 |
| LR decay epoch | 25 | 25 | 25 |
| Forward decay factor ($\alpha_{\mathrm{f}}$) | $^{63}/_{64}$ | - | - |
| Backward decay factor ($\alpha_{\mathrm{b}}$) | $^{1}/_{2}$ | - | - |

## A.3 Fully Connected

To test the Online Normalization technique on fully connected networks we use a three-layer dense network, 500+300 hidden units (3-layer NN, 500+300 HU, softmax, cross entropy, weight decay [29, 32]), with ReLU activation functions on the Fashion MNIST [30] classification task. The network is trained using the SGD optimizer and $L_2$ regularization. We consider three cases: without normalization, using Batch Normalization, Layer Normalization and Online Normalization. A learning rate sweep in the range 0.001 to 0.02 using a step size of 0.001 and the range 0.02 to 0.1 using a step size of 0.01 is performed for the network without normalization and with Batch Normalization. The networks using Layer Normalization and Online Normalization use the same hyperparameters found to be optimal for training when using Batch Normalization. A logarithmic sweep from $^{1}/_{2}$ to $^{8191}/_{8192}$ is performed to set the forward and backward decay factors. The optimum setting closely matched the hyperparameters used for ImageNet training. All hyperparameters are summarized in Table 5.

Table 5: Fully Connected Network Training Hyperparameters.

| | |
|---|---|
| Epoch | 10 |
| Batch size | 32 |
| Learning rate ($\eta$) | $4 \times 10^{-2}$ |
| $L_2$ constant ($\lambda$) | $10^{-4}$ |
| Forward decay factor ($\alpha_{\mathrm{f}}$) | 0.999 |
| Backward decay factor ($\alpha_{\mathrm{b}}$) | 0.99 |

## A.4 Recurrent Neural Network

For the recurrent network experiments we use single layer RNN and LSTM networks. The embedding and decoder are "tied" to share parameters as described in [33]. The networks are trained using SGD and $L_2$ regularization. The sequence length is selected uniformly in the range $[1, 128]$ to preclude the network from learning a sequence length. The recurrent networks are trained in three settings: using no normalization, Layer Normalization and Online Normalization. A linear sweep is done to set the learning rate (Table 7-8). A logarithmic sweep is used to set the forward and backward decay factors $\alpha_f$ and $\alpha_b$ (Table 7-8). All hyperparameters are summarized in Table 6.

## A.5 Gradient bias experiment

We used a simple network to quantify gradient bias for Batch Normalization (Section 3.2, Figure 3). The weights are held fixed to decouple learning rate changes from the bias. In our setup a single convolution layer with a normalizer is followed by ReLU feeding into a fully connected layer and

Table 6: Recurrent Network Training Hyperparameters.

| Recurrent Unit Type | RNN | | | LSTM | | |
|---|---|---|---|---|---|---|
| Normalization type | - | LN | ON | - | LN | ON |
| Learning rate ($\eta$) | 0.5 | 0.95 | 1.7 | 3.5 | 3.25 | 6.5 |
| Embedding size | | 200 | | | 200 | |
| Hidden state size | | 200 | | | 200 | |
| Epochs | | 40 | | | 25 | |
| Batch size | | 20 | | | 20 | |
| $L_2$ constant ($\lambda$) | | $10^{-6}$ | | | $10^{-6}$ | |
| Forward decay factor ($\alpha_f$) | | $^{16383}/_{16384}$ | | | $^{8191}/_{8192}$ | |
| Backward decay factor ($\alpha_b$) | | $^{127}/_{128}$ | | | $^{31}/_{32}$ | |

Table 7: RNN Network Hyperparameter Sweeps.

| Normalization type | - | LN | ON |
|---|---|---|---|
| Learning rate ($\eta$) | 0.5 | 0.95 | 1.7 |
| $\eta$ sweep range | 0.05 to 0.7 | 0.05 to 2 | 0.05 to 2 |
| $\eta$ sweep step size | 0.075 | 0.05 | 0.075 |
| Sweep range for $\alpha_f$ | $^{511}/_{512}$ to $^{32767}/_{32768}$ | | |
| Sweep range for $\alpha_b$ | $^{3}/_{4}$ to $^{4095}/_{4096}$ | | |

Table 8: LSTM Network Hyperparameter Sweeps.

| Normalization type | - | LN | ON |
|---|---|---|---|
| Learning rate ($\eta$) | 3.5 | 3.25 | 6.5 |
| $\eta$ sweep range | 2.5 to 10 | 1.25 to 5.75 | 1 to 10 |
| $\eta$ sweep step size | 0.5 | 1 | 0.5 |
| Sweep range for $\alpha_f$ | $^{511}/_{512}$ to $^{32767}/_{32768}$ | | |
| Sweep range for $\alpha_b$ | $^{3}/_{4}$ to $^{4095}/_{4096}$ | | |

softmax (Figure 13). We used the entire CIFAR-10 dataset to compute the ground truth gradient and compared it to the gradient resulting from batched computations using batch sizes in powers of two. The error shown represents the angle in degrees derived from cosine similarity of resulting gradients and the ground truth averaged over ten runs.

Figure 13: Network used to quantify gradient bias.

## A.6   Statistical Characterization of Experiment Reproducibility

The numerical values reported in Section 5 are median values for a set of runs. Figure 14 is a set of box-plots which statistically characterize the reproducibility of the experiments. Experiments with a single run are depicted using dashed lines. The run-to-run variability using Online Normalization is comparable to that of other normalizers.

The sensitivity of Online Normalization to decay rates when training ResNet20 on CIFAR10 is shown in Figure 15. For this fine-grained logarithmic sweep, the decay rates are expressed as the horizon of averaging $h = 1/(1 - \alpha)$. It shows that Online Normalization not highly sensitive to the chosen decay rate since the region of near-optimal performance is broad. This allows for coarser sweeps when generalizing the technique to different models and datasets.

Figure 14: Reproducibility.

Figure 15: Hyperparameter sweep.

# Appendix B  Gradient properties

The main part of the paper proved the expression of the gradient via projections (5) based on geometric considerations (Section 3.1). It is also possible to derive this property without geometry. Here is an alternative algebraic proof.

**Claim 1.** *In finite-dimensional spaces the backpropagation of the gradient of normalization (1) can be represented as a composition of two orthogonal projections:* $\vec{x}\,' = \frac{1}{\sigma}\left(I - P_{\vec{1}}\right)\left(I - P_{\vec{y}}\right)\vec{y}\,'$.

*Proof.* In the $N$-dimensional space transformation (1) becomes

$$\mu = \frac{1}{N}\sum_i x_i$$

$$\sigma^2 = \frac{1}{N}\sum_i (x_i - \mu)^2 \tag{13}$$

$$y_i = \frac{x_i - \mu}{\sigma}\,.$$

The derivatives of the mean and variance with respect to the $x_j$ are:

$$\frac{\partial \mu}{\partial x_j} = \frac{1}{N} \tag{14}$$

$$
\begin{aligned}
\frac{\partial \sigma}{\partial x_j} &= \frac{1}{2\sigma N}\sum_i \left[2(x_i - \mu)\left(\delta_{ij} - \frac{1}{N}\right)\right]\\
&= \frac{1}{N\sigma}\sum_i [(x_i - \mu)\delta_{ij}] - \frac{1}{N^2\sigma}\sum_i (x_i - \mu)\\
&= \frac{x_j - \mu}{N\sigma} - 0\\
&= \frac{y_j}{N}\,,
\end{aligned}
\tag{15}
$$

where $\delta_{ij}$ is the Kronecker delta function. The components of the Jacobian satisfy

$$
\begin{aligned}
J_{ij} \equiv \frac{\partial y_i}{\partial x_j} &= \frac{\left(\delta_{ij} - \frac{\partial \mu}{\partial x_j}\right)\sigma - (x_i - \mu)\frac{\partial \sigma}{\partial x_j}}{\sigma^2}\\
&= \frac{\left(\delta_{ij} - \frac{1}{N}\right) - y_i\frac{\partial \sigma}{\partial x_j}}{\sigma}\\
&= \frac{\left(\delta_{ij} - \frac{1}{N}\right) - \frac{y_i y_j}{N}}{\sigma}\\
&= \frac{(N\delta_{ij} - 1) - y_i y_j}{N\sigma}\,.
\end{aligned}
\tag{16}
$$

The $j$-th component of the gradient passing through normalization is

$$
\begin{aligned}
x'_j &= \frac{\partial L}{\partial x_j}\\
&= \sum_i \frac{\partial L}{\partial y_i}\frac{\partial y_i}{\partial x_j}\\
&= \frac{\sum_i \left(y'_i\left[(N\delta_{ij} - 1) - y_i y_j\right]\right)}{N\sigma}\\
&= \frac{Ny'_j - \sum_i y'_i - y_j\sum_i(y'_i y_i)}{N\sigma}\\
&= \frac{y'_j}{\sigma} - \frac{\sum_i y'_i}{N\sigma} - \frac{y_j\sum_i(y'_i y_i)}{N\sigma}\\
&= \frac{1}{\sigma}\left[y'_j - \frac{\sum_i y'_i}{N} - \frac{y_j\sum_i(y'_i y_i)}{N}\right]
\end{aligned}
\tag{17}
$$

and

$$\vec{x}\,' = \frac{1}{\sigma}\left[\vec{y}\,' - \frac{(\vec{y}\,', \vec{1})}{N}\vec{1} - \frac{(\vec{y}\,', \vec{y})}{N}\vec{y}\right]\,, \tag{18}$$

where $(\cdot, \cdot)$ is the inner product in $N$ dimensions.

Because $\|\vec{1}\|^2 = N$ and

$$
\begin{aligned}
\|\vec{y}\|^2 &= \sum_i y_i^2 \\
&= \sum_i \frac{N\left(x_i - \mu\right)^2}{\sum_j \left(x_j - \mu\right)^2} \\
&= N \ ,
\end{aligned}
\tag{19}
$$

we can express (18) in terms of the projections

$$
\begin{aligned}
\vec{x}\,' &= \frac{1}{\sigma} \left[ \vec{y}\,' - \frac{(\vec{y}\,', \vec{1})}{(\vec{1}, \vec{1})}\vec{1} - \frac{(\vec{y}\,', \vec{y})}{(\vec{y}, \vec{y})}\vec{y} \right] \\
&= \frac{1}{\sigma} \left( \mathrm{I} - \mathrm{P}_{\vec{1}} - \mathrm{P}_{\vec{y}} \right) \vec{y}\,' \ .
\end{aligned}
\tag{20}
$$

From this expression and because $\vec{y}$ is orthogonal to $\vec{1}$, we can see that resulting gradient $\vec{x}\,'$ is orthogonal to both $\vec{1}$ and $\vec{y}$.

Orthogonality of $\vec{y}$ and $\vec{1}$ also implies that $\mathrm{P}_{\vec{1}}\mathrm{P}_{\vec{y}} = 0$ and therefore

$$
\begin{aligned}
\vec{x}\,' &= \frac{1}{\sigma} \left( \mathrm{I} - \mathrm{P}_{\vec{1}} - \mathrm{P}_{\vec{y}} + \mathrm{P}_{\vec{1}}\mathrm{P}_{\vec{y}} \right) \vec{y}\,' \\
&= \frac{1}{\sigma} \left( \mathrm{I} - \mathrm{P}_{\vec{1}} \right) \left( \mathrm{I} - \mathrm{P}_{\vec{y}} \right) \vec{y}\,' \ .
\end{aligned}
\tag{21}
$$

$\square$

This proves equation (5) algebraically. Note that orthogonality conditions (4) follow from this representation.

## Appendix C   Weights and gradients equilibrium conditions

For the weight update shown in Figure 5 we have

$$
\begin{aligned}
|w|^2 - \left(\eta \mathbb{E}|w'|\right)^2 &= \left(|w| - \eta\lambda|w|\right)^2 \\
&= |w|^2 - 2\eta\lambda|w|^2 + \eta^2\lambda^2|w|^2
\end{aligned}
\tag{22}
$$

$$
\begin{aligned}
\left(\eta \mathbb{E}(|w'|)\right)^2 &= (2 - \eta\lambda)\eta\lambda|w|^2 \\
&\approx 2\eta\lambda|w|^2 \ .
\end{aligned}
\tag{23}
$$

Solving for equilibrium norm of the weights $|w|$ we get

$$
|w| = \sqrt{\frac{\eta}{2\lambda}}\mathbb{E}|w'|
\tag{24}
$$

and correspondingly

$$
\begin{aligned}
\frac{\Delta w}{|w|} &= \frac{\eta w'}{\sqrt{\frac{\eta}{2\lambda}}\mathbb{E}|w'|} \\
&= \sqrt{2\eta\lambda}\,\frac{w'}{\mathbb{E}\,|w'|}
\end{aligned}
\tag{25}
$$

matching equations (6) and (7).

# Appendix D Properties of Online Normalization

In this section we prove the properties of Online Normalization presented in Section 4. We focus on per-feature normalization in steps (8) and (11) and do not discuss layer scaling steps (9) and (10).

For simplicity in subsequent derivations we only consider the case of scalar samples. A generalization to multi-scalar samples is straightforward but clutters the equations. Under this simplification the forward process (8) can be rewritten as

$$y_t = \frac{x_t - \mu_{t-1}}{\sigma_{t-1}} \tag{26a}$$

$$\mu_t = \alpha\mu_{t-1} + (1-\alpha)x_t \tag{26b}$$

$$\sigma_t^2 = \alpha\sigma_{t-1}^2 + \alpha(1-\alpha)(x_t - \mu_{t-1})^2 . \tag{26c}$$

This process is a standard way to compute mean and variance of the incoming sequence $x$ via exponentially decaying averaging:

$$\mu_t = (1-\alpha)\sum_{j=0}^{t}\alpha^{t-j}x_j \tag{27}$$

$$\sigma_t = (1-\alpha)\sum_{j=0}^{t}\alpha^{t-j}(x_j - \mu_t)^2 . \tag{28}$$

We start with an observation that the computation of the mean in (26) can be equivalently performed as a control process:

**Claim 2.** *Control process*

$$\begin{aligned}\hat{y}_t &= x_t - (1-\alpha)\varepsilon_{t-1}\\ \varepsilon_t &= \varepsilon_{t-1} + \hat{y}_t.\end{aligned} \tag{29}$$

*is equivalent to estimator process (26b)*

$$\begin{aligned}\hat{y}_t &= x_t - \mu_{t-1}\\ \mu_t &= \alpha\mu_{t-1} + (1-\alpha)x_t\end{aligned} \tag{30}$$

*with the accumulated control error $\varepsilon_t$ proportional to the running mean $\mu_t$*

$$\mu_t = (1-\alpha)\varepsilon_t . \tag{31}$$

*Proof.* The equivalence of the first lines is obvious. From (29) and (31) we also have

$$\begin{aligned}\mu_t &= (1-\alpha)\varepsilon_t\\ &= (1-\alpha)(\varepsilon_{t-1} + \hat{y}_t)\\ &= \mu_{t-1} + (1-\alpha)(x_t - (1-\alpha)\varepsilon_{t-1})\\ &= \mu_{t-1} + (1-\alpha)(x_t - \mu_{t-1})\\ &= \alpha\mu_{t-1} + (1-\alpha)x_t ,\end{aligned} \tag{32}$$

which matches (30). $\square$

To proceed we make an assumption that the input to the normalizer is bounded:

**Assumption 1.** *We assume that inputs $x$ are bounded: $|x_t| < C_x \quad \forall t$.*

**Claim 3.** *Under this assumption, the accumulated output of process (30) is uniformly bounded by*

$$\left|\sum_{j=0}^{t}\hat{y}_j\right| < \frac{1}{1-\alpha}C_x \quad \forall t . \tag{33}$$

*Proof.* Second line of (29) implies that

$$\sum_{j=0}^{t} \hat{y}_j = \varepsilon_t \ . \tag{34}$$

From representation (27) and equality (31) we have

$$
\begin{aligned}
\left| \sum_{j=0}^{t} \hat{y}_j \right| &= |\varepsilon_t| \\
&= \frac{|\mu_t|}{1-\alpha} \\
&= \left| \sum_{j=0}^{t} \alpha^{t-j} x_j \right| \\
&< C_x \sum_{j=0}^{\infty} \alpha^j \\
&= \frac{C_x}{1-\alpha} \ .
\end{aligned}
\tag{35}
$$

$\square$

Process (26) is identical to process (30) except scaling with $\sigma$

$$y_t = \frac{\hat{y}_t}{\sigma_{t-1}} \ . \tag{36}$$

To extend the result of Claim 3 to (26) we assume that there is nonzero variability in the input.

**Assumption 2.** *Variance of the input stream $x$ computed via exponentially decaying averaging (26c, 28) is uniformly bounded away from zero after initial $N$ steps:*

$$\sigma_t^2 > C_\sigma^2 > 0 \quad \forall t \geq N \ . \tag{37}$$

Note that this assumption only requires that there is sufficient variability in the input for successful normalization. The first $N$ steps correspond to the warmup of the process when the approximated statistics may experience high variability.

**Claim 4.** *Arbitrarily long accumulated sum of output of the process (26) starting with time step N is uniformly bounded by*

$$\left| \sum_{j=N+1}^{t} y_j \right| < \frac{1}{1-\alpha} \frac{2C_x}{C_\sigma} \quad \forall t \ . \tag{38}$$

*Proof.* From the bound (35) and equivalence (36) for any $t$ have

$$
\begin{aligned}
\left| \sum_{j=N+1}^{t} y_j \right| &= \left| \sum_{j=N+1}^{t} \frac{\hat{y}_j}{\sigma_{t-1}} \right| \\
&< \frac{1}{C_\sigma} \left| \sum_{j=N+1}^{t} \hat{y}_j \right| \\
&\leq \frac{1}{C_\sigma} \left( \left| \sum_{j=0}^{N} \hat{y}_j \right| + \left| \sum_{j=0}^{t} \hat{y}_j \right| \right) \\
&< \frac{1}{C_\sigma} \frac{2C_x}{1-\alpha} \ .
\end{aligned}
\tag{39}
$$

$\square$

This uniform bound implies that the average of the normalized stream $y_j$ generated by (26) asymptotically approaches zero as the window of averaging increases.

**Claim 5.** *After initial N steps (Assumption 2), the output $y$ generated by generated by (26) satisfies*

$$\lim_{t \to \infty} \mu_t(y) \equiv \lim_{t \to \infty} \left( \frac{1}{t} \sum_{j=N+1}^{N+t} y_j \right) = 0 \; , \tag{40}$$

We can construct a similar result for the variance of $y$.

**Claim 6.** *Output $y$ generated by (26) satisfies*

$$\lim_{t \to \infty} \sigma_t^2(y) \equiv \lim_{t \to \infty} \left( \frac{1}{t} \sum_{j=N+1}^{N+t} (y_j - \mu_t(y))^2 \right) = \frac{1}{\alpha} \tag{41}$$

*Proof.* Based on the equality $\sigma^2(y) = \mu(y^2) - \mu(y)^2$ and Claim 5 we observe that

$$
\begin{aligned}
\lim_{t \to \infty} \sigma_t^2(y) &= \lim_{t \to \infty} \left( \frac{1}{t} \sum_{j=N+1}^{N+t} y_j^2 \right) - \lim_{t \to \infty} \left( \frac{1}{t} \mu_t(y) \right)^2 \\
&= \lim_{t \to \infty} \left( \frac{1}{t} \sum_{j=N+1}^{N+t} \frac{(x_j - \mu_{j-1})^2}{\sigma_{j-1}^2} \right) \; .
\end{aligned}
\tag{42}
$$

From (26c) we have $(x_j - \mu_{j-1})^2 = (\sigma_j^2 - \alpha \sigma_{j-1}^2)/(\alpha(1-\alpha))$, and therefore

$$
\begin{aligned}
\lim_{t \to \infty} \sigma_t^2(y) &= \lim_{t \to \infty} \left( \frac{1}{t} \sum_{j=N+1}^{N+t} \frac{\sigma_j^2 - \alpha \sigma_{j-1}^2}{\alpha(1-\alpha)\sigma_{j-1}^2} \right) \\
&= \lim_{t \to \infty} \left( \frac{1}{t} \sum_{j=N+1}^{N+t} \frac{\sigma_j^2 - \sigma_{j-1}^2 + (1-\alpha)\sigma_{j-1}^2}{\alpha(1-\alpha)\sigma_{j-1}^2} \right) \\
&= \lim_{t \to \infty} \left( \frac{1}{t} \sum_{j=N+1}^{N+t} \frac{\sigma_j^2 - \sigma_{j-1}^2}{\alpha(1-\alpha)\sigma_{j-1}^2} \right) + \frac{1}{\alpha} \\
&= \frac{1}{\alpha} \; .
\end{aligned}
\tag{43}
$$

$\square$

Note that the resulting asymptotic variance approaches 1 as $\alpha$ approaches 1 (in our experiments $\alpha \approx 0.999$). Additionally, any fixed asymptotic variance in all features will be absorbed in subsequent layer scaling bringing resulting variance to 1.

Combined, the previous two claims prove the following property.

**Property 1.** *Output $y$ generated by the forward pass of Online Normalization (26) is asymptotically mean zero and unit variance.*

Now we analyze the stability of the algorithm with respect to imperfect estimates $\mu$ and $\sigma$.

**Claim 7.** *Derivatives of the output $y$ generated by (26) with respect to $\mu$ and $\sigma$ are bounded.*

*Proof.* We first observe that under previous assumptions $y$ is bounded

$$
\begin{aligned}
|y_t| &= \left| \frac{x_t - \mu_{t-1}}{\sigma_{t-1}} \right| \\
&\leq \left| \frac{1}{\sigma_{t-1}} \right| (|x_t| + |\mu_{t-1}|) \\
&< \frac{2C_x}{C_\sigma} \equiv C_y \; .
\end{aligned}
\tag{44}
$$

The derivatives of $y$ are

$$\left| \frac{\partial y_t}{\partial \mu_{t-1}} \right| = \left| \frac{1}{\sigma_{t-1}} \right|$$
$$< \frac{1}{C_\sigma} \tag{45}$$

and

$$\left| \frac{\partial y_t}{\partial \sigma_{t-1}} \right| = \left| \frac{x_t - \mu_{t-1}}{\sigma_{t-1}^2} \right|$$
$$= \left| \frac{y_t}{\sigma_{t-1}} \right| \tag{46}$$
$$< \frac{C_y}{C_\sigma} \ .$$

$\square$

Because normalized output $y$ is a continuous function of running estimates of $\mu$ and $\sigma$ with bounded derivatives, errors in the estimates have a bounded effect on the result.

**Property 2.** *The deviation of the output of Online Normalization (26) from normal distribution is a Lipschitz function with respect to errors in estimates of mean and variance of its input.*

In particular, it means that with sufficiently small learning rate, the normalization process is guaranteed to produce generate outputs with mean and variance arbitrarily close to zero and one even when the network parameters are changing.

Now we turn our attention to the corresponding backward pass (11-12), which in the case of single scalar per sample becomes

$$\tilde{x}'_t = y'_t - (1-\alpha)\varepsilon_{t-1}^{(y)} y_t$$
$$\varepsilon_t^{(y)} = \varepsilon_{t-1}^{(y)} + \tilde{x}'_t y_t \tag{47}$$

and

$$x'_t = \frac{\tilde{x}'_t}{\sigma_{t-1}} - (1-\alpha)\varepsilon_{t-1}^{(1)}$$
$$\varepsilon_t^{(1)} = \varepsilon_{t-1}^{(1)} + x'_t \ . \tag{48}$$

We can formulate the counterpart of Claim 2 for this process. for (47) is

**Claim 8.** *Control process (47) is equivalent to estimator process*

$$\tilde{x}'_t = y'_t - \mu_{t-1}^{(y)} y_t$$
$$\mu_t^{(y)} = (1 - (1-\alpha)y_t^2)\mu_{t-1}^{(y)} + (1-\alpha)y'_t y_t \tag{49}$$

*with*

$$\mu_t^{(y)} = (1-\alpha)\varepsilon_t^{(y)} \ . \tag{50}$$

*Proof.* Similarly to the proof of Claim 2 we have

$$\begin{aligned}
\mu_t^{(y)} &= (1-\alpha)\varepsilon_t^{(y)} \\
&= (1-\alpha)(\varepsilon_{t-1}^{(y)} + \tilde{x}'_t y_t) \\
&= \mu_{t-1}^{(y)} + (1-\alpha)\left(y'_t - (1-\alpha)\varepsilon_{t-1}^{(y)} y_t\right) y_t \\
&= \mu_{t-1}^{(y)} + (1-\alpha)\left(y'_t - \mu_{t-1}^{(y)} y_t\right) y_t \\
&= (1 - (1-\alpha)y_t^2)\mu_{t-1}^{(y)} + (1-\alpha)y'_t y_t \ ,
\end{aligned} \tag{51}$$

which matches (49).

$\square$

**Assumption 3.** *The incoming gradient $y_t'$ is bounded:*

$$y_t' < C_{y'} \quad \forall t \tag{52}$$

*and that exponentially decaying average of normalized output $y_t^2$ is bounded away from zero:*

$$(1-\alpha)\sum_{j=0}^{t}\alpha^{t-j}y_t^2 > C_{y^2} > 0 \quad \forall t > N . \tag{53}$$

The last condition is natural given that $y_t$ is the result of forward normalizations and we have shown that it is asymptotically mean zero and $1/\alpha$ variance.

**Assumption 4.** *The decay factor $\alpha$ for the backward pass is sufficiently close to one to satisfy*

$$C_y > \frac{1}{1-\alpha} . \tag{54}$$

**Claim 9.** *Error accumulator $\varepsilon_t^{(y)}$ in (47) is bounded.*

*Proof.* Because of the equivalency shown in Claim 8 it is sufficient to prove the statement only for $\mu_t^{(y)}$ in (49). For $t > N$ we have

$$\mu_t^{(y)} = (1-(1-\alpha)y_t^2)\mu_{t-1}^{(y)} + (1-\alpha)y_t'y_t$$

$$\mu_t^{(y)} = (1-(1-\alpha)y_t^2)\left[(1-(1-\alpha)y_{t-1}^2)\mu_{t-2}^{(y)} + (1-\alpha)y_{t-1}'y_{t-1}\right] + (1-\alpha)y_t'y_t$$

$$= \ldots \tag{55}$$

$$= (1-\alpha)\sum_{k=0}^{t}\left[\prod_{j=0}^{k-1}\left(1-(1-\alpha)y_{t-j+1}^2\right)\right]y_{t-k}'y_{t-k} ,$$

and

$$|\mu_t^{(y)}| < (1-\alpha)NC_yC_{y'} + (1-\alpha)C_yC_{y'}\sum_{k=0}^{t-N}\left[\prod_{j=0}^{k-1}\left(1-(1-\alpha)y_{t-j+1}^2\right)\right] . \tag{56}$$

If individual values of $y_t^2$ were bounded below, the summation would be done over a geometric progression converging to a bounded value. But individual values of $y_t^2$ can be zero so we cannot directly bound the sum by a converging geometric series. Instead, we'll use the property that the exponentially averaged $y_t^2$ is bounded away from zero to show that it implies that the arithmetic average of any sufficiently long consecutive sequence of $y_t^2$ is bounded away from zero and use that to bound $\mu^{(y)}$.

First we notice that we can replace the last term in (56) by a power of arithmetic average using the convexity property

$$\prod_{j=0}^{k-1}(1-\alpha_j) \leq \left(1 - \frac{1}{k}\sum_{j=0}^{k-1}\alpha_j\right)^k \quad \text{if} \quad \alpha_j \quad \forall j \tag{57}$$

that can be proven inductively starting with $k = 2$. Then, after substituting $\alpha_j \leftarrow (1-\alpha)y_{t-j+1}^2$, inequality (56) becomes

$$|\mu_t^{(y)}| < (1-\alpha)NC_yC_{y'} + (1-\alpha)C_yC_{y'}\sum_{k=0}^{t-N}\left(1-(1-\alpha)\left(\frac{1}{k}\sum_{j=0}^{k-1}y_{t-j}^2\right)\right)^k . \tag{58}$$

Finally, if we show that the averages in (58) are bounded from below by a nonzero positive constant then the resulting geometric sum with the fixed base less than one will be bounded.

For $\alpha < 1$ the series $(1-\alpha)\sum\alpha^k$ is converging and therefore we can find $K$ such that the tail of this series is less than a fixed value $C_{y^2}/2C_{y+}$:

$$(1-\alpha)\sum_{k=K}^{\infty}\alpha^k < \frac{C_{y^2}}{2C_{y+}} . \tag{59}$$

This is true when

$$\alpha^K < (1-\alpha)\frac{C_{y^2}}{2C_y}$$

$$K \log \alpha < \log \frac{(1-\alpha)C_{y^2}}{2C_y} \tag{60}$$

$$K = \left\lceil \log \frac{(1-\alpha)C_{y^2}}{2C_y} \middle/ \log \alpha \right\rceil .$$

Combining (54) and (59) for all $n > N$ we get a lower bound for the top $K$ terms in (53)

$$
\begin{aligned}
(1-\alpha) \sum_{k=t-K+1}^{t} \alpha^{t-k} y_k^2 &= (1-\alpha) \sum_{k=0}^{t} \alpha^{t-k} y_k^2 - (1-\alpha) \sum_{k=0}^{t-K} \alpha^{t-k} y_k^2 \\
&> C_{y^2} - (1-\alpha)C_y \sum_{k=K}^{\infty} \alpha^k \\
&> C_{y^2} - \frac{C_{y^2}}{2} \\
&= \frac{C_{y^2}}{2} .
\end{aligned} \tag{61}
$$

Then for all $t > N$ we can bound from below the arithmetic average of the $K$ corresponding terms of $y$.

$$
\begin{aligned}
\frac{1}{K} \sum_{k=0}^{K-1} y_{t-k}^2 &> \frac{1}{\alpha^{K-1}} \sum_{k=0}^{K-1} \alpha^k y_{t-k}^2 \\
&> \frac{C_{y^2}}{2(1-\alpha)\alpha^{K-1}} \equiv C_{\bar{y}} > 0 .
\end{aligned} \tag{62}
$$

That shows that after the first $N$ terms, the average of any consecutive $K$-sequence of $y$ exceeds a fixed constant. For any $t$ and $K' > K$ we can apply this property to $\lfloor K'/K \rfloor$ $K$-chunks to get

$$
\begin{aligned}
\frac{1}{K'} \sum_{k=0}^{K'-1} y_{t-k}^2 &> \left\lfloor \frac{K'}{K} \right\rfloor \frac{K}{K'} C_{\bar{y}} \\
&> \frac{C_{\bar{y}}}{2} .
\end{aligned} \tag{63}
$$

Combining (58) and (63) we get the bound

$$
\begin{aligned}
|\mu_t^{(y)}| &< (1-\alpha)(N+K)C_{y'}C_y + (1-\alpha)C_{y'}C_y \sum_{k=K}^{t-N} \left(1 - (1-\alpha)\left(\frac{1}{k}\sum_{j=0}^{k-1} y_{t-j}^2\right)\right)^k \\
&< (1-\alpha)(N+K)C_{y'}C_y + (1-\alpha)C_{y'}C_y \sum_{k=K}^{t-N} \left(1 - (1-\alpha)\frac{C_{\bar{y}}}{2}\right)^k \\
&< (1-\alpha)(N+K)C_{y'}C_y + (1-\alpha)C_{y'}C_y \frac{2}{(1-\alpha)C_{\bar{y}}} \\
&= C_{y'}C_y \left((1-\alpha)(N+K) + \frac{2}{C_{\bar{y}}}\right) \equiv C_{\mu^y} ,
\end{aligned} \tag{64}
$$

and because of the equivalency (50) between $\mu_t^{(y)}$ and $\varepsilon_t^{(y)}$

$$|\varepsilon_t^{(y)}| < \frac{C_{\mu^y}}{1-\alpha} \equiv C_{\varepsilon^y} . \tag{65}$$

$\square$

**Claim 10.** $\tilde{x}'_t$ in process (47), (49) is uniformly bounded.

*Proof.* From (49) and bounds on

$$
\begin{aligned}
|\tilde{x}'_t| &= |y'_t - \mu^{(y)}_{t-1} y_t| \\
&\leq |y'_t| + |\mu^{(y)}_{t-1}||y_t| \\
&= C_{y'} + C_{\mu^y} C_y \ .
\end{aligned}
\tag{66}
$$

$\square$

The second stage of the backward pass (48) is the same is the process (29) with input $\tilde{x}'_t/\sigma_{t-1}$ that is bounded:

$$
\left|\frac{\tilde{x}'_t}{\sigma_{t-1}}\right| < \frac{C_{y'} + C_{\mu^y} C_y}{C_\sigma} \ .
\tag{67}
$$

We can reuse the earlier results to conclude that both the output of (48) $x'_t$ and accumulated error $\varepsilon^{(1)}_t = \sum x'_t$ are bounded:

$$
|x'_t| < C_{x'}
\tag{68}
$$

and

$$
|\varepsilon^{(1)}_t| < C_{\varepsilon^1} \ .
\tag{69}
$$

These observations together with (65) can be restated as properties.

**Property 3.** *The backward pass of Online Normalization (11)-(12) generates uniformly bounded gradients $x'_t$.*

**Property 4.** *Accumulated errors $\varepsilon^{(y)}_t$ and $\varepsilon^{(1)}_t$ that track deviations from orthogonality conditions (5) in Onine Normalization (11)-(12) are bounded.*

## Appendix E   Emulation of Online Normalization on GPU

While Online Normalization offers a normalization technique that does not rely on batching, some hardware architectures benefit from batched execution of compute-intensive linear operations. For fast GPU execution we reformulated the algorithm to operate on tensors with the batch dimension and still generate results equivalent to true online processing. Of course this forces the weight updates to be performed on batch boundaries, which the original algorithm does not require.

Let's assume that we are computing the exponentially decaying mean of a sequence of inputs $x_t$ (26b)

$$
\mu_t = \alpha \mu_{t-1} + (1-\alpha)x_t \ ,
\tag{70}
$$

which is equivalent to (27)

$$
\begin{aligned}
\mu_t &= (1-\alpha) \sum_{j=0}^{t} \alpha^{t-j} x_j \\
&= (1-\alpha) \sum_{j=0}^{t} \alpha^j x_{t-j} \ .
\end{aligned}
\tag{71}
$$

We also assume that inputs $x_t$ arrive in groups of $n$ elements

$$
\begin{aligned}
X_{t-n} &= (x_{t-n}, \ldots, x_{t-1}) \\
X_t &= (x_t, \ldots, x_{t+n-1}) \ ,
\end{aligned}
\tag{72}
$$

where $X_{t-n}$ is a previously processed group with resulting values

$$
M_{t-n} = (\mu_{t-n}, \ldots, \mu_{t-1})
\tag{73}
$$

matching (71) and $X_i$ is the current batch that we need to process and generate

$$
M_t = (\mu_t, \ldots, \mu_{t+n-1}) \ .
\tag{74}
$$

We will use the superscript to refer to a specific element of the the group

$$M_t^l \equiv \mu_{t+l} = (1 - \alpha) \sum_{j=0}^{t+l} x_{t+l-j} \alpha^j . \tag{75}$$

We will also use a $n$-vector of powers of $\alpha$

$$A = \left(1, \alpha, \ldots, \alpha^{n-1}\right) \tag{76}$$

and a $(2n - 1)$-long concatenation of two adjacent $X$ batches (with the very first element removed):

$$X_{t-n,i} = (x_{t-n+1}, \ldots, x_t, \ldots, x_{t+n-1}) . \tag{77}$$

Multiplying previously computed batch by $\alpha^n$ we get

$$\begin{aligned} \alpha^n M_{t-n}^l &= \alpha^n \mu_{t-n+l} \\ &= (1 - \alpha) \sum_{j=0}^{t-n+l} x_{t-n+l-j} \alpha^{j+n} \\ &= (1 - \alpha) \sum_{j=n}^{t+l} x_{t+l-j} \alpha^j . \end{aligned} \tag{78}$$

This matches our target expression (75) except the summation starts from $n$ instead of zero. We can cover the missing summation range by applying a 1D convolution with filter (76) to (77):

$$\begin{aligned} (X_{t-n,i} \circledast A)^l &= \sum_{j=0}^{n} X_{t-n,t}^{l+n-j} A^j \\ &= \sum_{j=0}^{n} x_{t+l-j} \alpha^j . \end{aligned} \tag{79}$$

Therefore we can generate target values (75) as

$$\begin{aligned} M_t^l &= \mu_{t+l} \\ &= (1 - \alpha) \sum_{j=0}^{t+l} x_{t+l-j} \alpha^j \\ &= \alpha^n M_{t-n}^l + (1 - \alpha) \left(X_{t-n,t} \circledast A\right)^l . \end{aligned} \tag{80}$$

The resulting group-level expression is

$$M_t = \alpha^n M_{t-n} + (1 - \alpha) \left(X_{t-n,t} \circledast A\right) , \tag{81}$$

where $M_{t-n}$ is the previously computed batch of results, $X_{t-n,t}$ is the concatenation of the previous and current batches of $x$ (without the very first element), $A$ is the vector of $n$ powers of $\alpha$, and $\circledast$ is the 1D convolution. In the limit case of $n = 1$ this expression matches the origingal method. With $n > 1$ and $X$ and $M$ initialized to zero tensors the resulting procedure will match (in exact arithmetic) the values of the streaming process (26b) with standard initialization.

The generalization of this method to the computation of variance (26c) and to the procedure (47-48) in the backward pass can be found in the accompanying code [3].

## Appendix F   Hyperparameter scaling rules

In our studies we performed experiments with different batch sizes. For momentum training

$$\begin{aligned} \nu &= \mu\nu + (1 - \mu)g \\ w &= w - \eta\nu , \end{aligned} \tag{82}$$

we applied scaled the learning rate linearly with batch size $b$:

$$\eta_{new} = \frac{b_{new}}{b_{old}} \eta_{old}, \tag{83}$$

while keeping the weight decay parameter unchanged. This effectively leads to a square root scaling rule for training (Section 3.4).

To scale the momentum $\mu$ in (82) we equate per-sample decay

$$\mu_{new}^{\frac{1}{b_{new}}} = \mu_{old}^{\frac{1}{b_{old}}} , \tag{84}$$

which results in

$$\mu_{new} = \mu_{old}^{\frac{b_{new}}{b_{old}}} . \tag{85}$$

Note that some deep learning frameworks implement momentum as outlined in [34]:

$$\begin{aligned} \nu &= \mu\nu + g \\ w &= w - \eta\nu , \end{aligned} \tag{86}$$

This is equivalent to (82) except the gradient is not multiplied by $(1 - \mu)$. To apply hyperparameter updates to momentum optimizers implemented by these deep learning frameworks, we apply another scale to the learning rate:

$$\eta_{new}^* = \frac{1 - \mu_{new}}{1 - \mu} \eta_{new} . \tag{87}$$