[Reviews · NeurIPS 2019]

Reviewer 1



The paper is well motivated and quite clear. I like the distinction between statistical, functional and heuristics methods of normalization. Also, investigating normalization techniques that do not rely on mini-batch statistics is an important research direction. I have however a few remarks concerning ON: 1) How does it compares to Batch Renormalization (BRN)? Both methods rely on running averages of statistics, so I think it would be fair to clearly state what are the differences between the two methods and to thoroughly compare against it in the experimental setup, especially because BRN introduces 1 extra hyper-parameter, while one need to tune 2 of them in ON. 2) How difficult is it to tune both decay rates hyper-parameters? Is ON robust to poor choices of decay rates? It would be nice to have an experiment showing their impact on performances (maybe a heatmap?). 3) It is stated several times that "ON has the best validation performances". However, without proper mean and standard deviations of the results, it is difficult to infer how good it is compared to other methods (especially when the difference in results is so small). So having the standard deviations on each curves and in each table would be more compelling. Also, I think it is more typical to compare models using accuracy rather than loss, because it is the metric we actually care about (and sometimes the loss is not a good proxy for accuracy, as one can see on the ImageNet results). And here are a few remarks on the experimental setup: CIFAR10: - What is the Batch Size used for online normalization? 1 or 128 like in BN (footnote "a" in table 2 confuses me)? - The "validation" set you used is actually the test set of CIFAR10. Validation should be performed on a fraction of the training set (usually the last 5000 examples), and final results should be reported using the test set. Language Modelling: In Figure 11, Layer Normalization performs as poorly as no normalization. After looking at your implementation, it seems that you implemented LN wrongly: the gates should be computed like this: gates = LN(Wx) + LN(Uh) + b rather than: gates = LN(Wx + Uh + b) After Author Response: Thanks your for your response and clarifications! Here are my last comments: - I appreciated the comparison with BRN, and I really do think it makes a strong case for your paper. - Given the rather big overlap in Figure 4 (at least in the feed-forward case), I would suggest to change the claims: "ON has the best validation performances" to something more subtle: "ON performs similarly as BN, while removing the dependency on the mini-batch" (or similar). That said, I increase my score to a 7.

Reviewer 2



The paper "Online Normalization for Training Neural Networks" proposes a novel technique for normalization when training neural networks. While classical batch normalization results in biased gradients even for moderate batch sizes, online normalization introduces exponential decaying averages of online samples, which presents interesting formal properties. The paper appears theoretically well-sounded. The problem is important in the field of neural networks. Experimental results are pretty convincing. My main concern is about the presentation of the paper. I had a really hard time with it, as a lot of details are omitted and contributions not enough highligthed. Authors should take a more pedagogical approach, where they present in more details classical normalization techniques, so that the reader can have everything in mind, in a formal version that allows him to better apprehend the notations used in the paper. Several parts are particularly hard to follow, in particular 2.2, 2.3 and 2.4, where plots given in the corresponding figures are not sufficiently detailled. More discussions about results would also be needed. Also I am still not sure whether the mu and sigma are computed over samples of the current batch for each feature or they are computed over all features. For me it's the former but some sentences make me rather uncertain about this (particularly sentence 187 about image samples). Please clarify. I have read the author response and my opinion remains the same.

Reviewer 3



The authors propose a new normalization method, called Online Normalization. The new method drops the dependency on batch size and allows neurons to locally estimate forward mean and variance. Furthermore, Online Normalization also implements a control process on backward pass to ensure that back-propagated gradients stay within a bounded distance of true gradients. Experimental results show that Online Normalization can reduce memory usage largely with the matching accuracy comparing to Batch Normalization. Strength: 1. This work drops the dependency on batch size, leading to lower memory usage and wider applications. 2. The authors resolve the biased problem in Batch Normalization and introduce an unbiased technique for computing the gradient of normalized activations. 3. Experiments are solid and convincing. The experimental settings are described clearly. 4. The paper is written clearly. The structure is well-organized and It is easy to read and understand. Weakness 1. Online Normalization introduces two additional hype-parameters: forward and backward decay factors. The authors use a logarithmic grid sweep to search the best factors. This operation largely increases the training cost of Online Normalization. Question: 1. The paper mentions that Batch Normalization has the problem of gradient bias because it uses mini-batch to estimate the real gradient distribution. In contrast, Online Normalization can be implemented locally within individual neurons without the dependency on batch size. It sounds like that Online Normalization and Batch Normalization are two different ways to estimate the real gradient distribution. I am confused why Online Normalization is unbiased and Batch Normalization is biased. ** I have read other reviews and the author response. I will stay with my original score. **

Reviewer 4



The paper proposes to enable single-example normalization that, for each layer in expectation, is equivalent to batchnorm. For this purpose, the paper observes that the normalization statistics can be estimated as moving averages, and the gradients through them could be estimated as moving averages E[g] (g is the backprop input) for the mean subtraction and E[gx] for the divisive normalization. The proposed approach adds to this the division by the total norm of the activations (akin to layer normalization), ostensibly to combat the explosion / vanishing of activations. Positive results are shown on several image tasks with ResNets (for classification) and U-Net (segmentation), as well as RNN and LSTM on PTB. Overall the paper makes sense, and the results are encouraging. However, there are several significant issues that I think make the paper less strong than it could be: - All the image results are shown using models that, as has been shown, can be trained without any normalization. Specifically, the ResNet architecture has been hypothesized to enable better gradient propagation, could be initialized in a way that makes the normalization unnecessary, and by this token could be more forgiving to the choice of normalization scheme used. I think the results would be stronger if they also used deep non-residual architectures, such as Inception, to demonstrate that the proposed scheme performs well even in the absence of skip connections. - The layer scaling step in Eqn. (9) means that the proposed architecture requires inference-time normalization (similar to LN and GN). I think this should be made more explicit. Regarding the reason for this step, the paper suggests that it is to combat the vanishing / exploding activations. I would like to see this statement substantiated. Specifically, I would like to know that indeed this is to bound the activations, rather than to e.g. reduce the bias / variance of gradients. For instance, could it be that the proposed scheme without the layer scaling would likely produce gradients w.r.t. the weights that are not orthogonal to the weights, and the layer scaling fixes the issue? A simple way to verify this would be to consider other activation-bounding techniques, such as clipping the activations. Some more-minor comments: - The paper claims to compute unbiased gradient estimates -- I am not sure that they are unbiased. The computation at any given layer may be unbiased, but the product of these may not be since the Jacobians of different layers are not independent. - I did not understand the "freezing" part in lines 93-98. What is meant by this? - In Eqns (11b) and (12b), is there a multiplier \alpha_g missing before the \epsilon on the RHS? - Lines 205-207, I am not sure that the gradient scale invariance can be invoked here for the cases where the normalization occurs in one of several parallel branches (which happens in e.g. ResNets). - Overall, the paper is rather clear. However, I did not find Figure 1 to contribute to this clarity -- I did not make a significant effort to understand it though, as I assume it illustrates what is already very clear in the text. - The projections performed when backpropping through batchnorm are well known -- e.g. second half of sec. 3 of https://arxiv.org/pdf/1702.03275.pdf. - I would love to see the optimal values of \alpha_f and \alpha_g for at least some of the tasks in the paper text, rather than the appendix. Overall I think the paper as it is could be of value to practitioners, but it could be made more so.

[Author Response · NeurIPS 2019]



Figure 1: BRN vs ON (CIFAR10/ResNet20).

Figure 2: LSTM training.

Figure 3: Hyperparameter sweep (CIFAR10/ResNet20).

Figure 4: Run-to-run variability.

Dear Reviewers R1, R2, and R3: Thank you for your comments and suggestions to improve our paper.

**Comparison with Batch Renormalization (BRN) [R1]:** Online Normalization (ON) processes all data without
batches, and differentiates through its estimation process. In contrast, BRN's statistical estimates are based on batches
and it does not differentiate through its estimation. To counter instability, BRN adds two threshold hyperparameters to
constrain its gradients to a neighborhood of the current batch's statistics. Like Instance Normalization, BRN cannot be
used for dense layers at batch-1, and performs poorly on convolutions. The BRN paper reports 2% accuracy reduction
for ImageNet at batch-4. We trained CIFAR-10 with BRN with even smaller batches (Figure 1), using hyperparameters
and schedule based on the BRN paper's guidelines. Notice the sharp reduction in accuracy at batch-1.

**Note on Batch Size [R1]:** ON never requires batching. In Appendix E (supplemental materials) we present a method
to run ON efficiently on GPUs. This mode preserves causality: ON still operates exactly as it did at batch-1, whereas
the weights are updated at batch-N boundaries. This also allowed us to use the same hyperparameters (learning rate and
momentum) of Batch Normalization for comparison.

**Layer Norm (LN) LSTM [R1]:** Our network is based on TensorFlow's LayerNormBasicLSTMCell. We compared it
with the original version of Ba et al. After tuning its hyperparameters, we observed that it performs worse (Figure 2).

**ON Hyperparameters [R1, R3]:** ON removes the batch size parameter and introduces two decay rate parameters. For
small scale experiments we characterized the sensitivity of ON to the decay rates expressed as half-life of averaging
$h = 1/(1-\alpha)$. Because the region of near-optimal performance is broad (Figure 3), we reused the same hyperparameter
settings from CIFAR10 on ImageNet without any tuning. We will include this figure in the paper's appendix.

**CIFAR Validation [R1]:** We selected hyperparameter values in the middle of the near-optimal plateau (Figure 3,
green). Note, this is not the best value observed in the sweep. We agree that using a holdout set is the best practice.
Although we can not "undo" our experiment and forget the hyperparameter values, if we did run this sweep using a
holdout set, the same plateau would still be discernible. Its center point will stable to the additional noise caused by the
smaller dataset. The hyperparamters chosen for CIFAR also produced good results on ImageNet without any tuning.

**Error Bars [R1]:** Error bars reduced the plots' readability. Statistical characterization is present with the supplementary
code (file experiments/ExpReproducibility.md). The run-to-run variability using ON is comparable to that of other
normalizers. We will plot the full set of characterizations (example in Figure 4) and include them in the appendix.

**Computation of $\mu$ and $\sigma$ [R2]:** We compute $\mu$ and $\sigma$ per feature for each sample (line 188) and never over a batch.
The layer scaling step is the only computation that looks across features within each sample (line 200).

**Gradient Bias [R3]:** Estimates of $\mu$ and $\sigma$ can be made arbitrarily accurate with appropriate choices of the learning
rate and hyperparameters that depend on the Lipschitz constant of the loss surface. In this case the expected value of the
gradient will converge to the true gradient (Appendix D), therefore the gradients produced using ON are unbiased. In
contrast, for batch normalization there is an inherent bias once the batch size is fixed. We provide a simple example of
this in lines 126 - 130 and empirically show this effect for small batches (Paper Figure 2).

[Meta-Review · NeurIPS 2019]

The authors propose a new normalization technique for training deep networks called online normalization, as an alternative to batch normalization, providing both theoretical analyses and experimental results for the proposed approach. The topic is likely to be of broad interest to the NeurIPS audience given the prevalence of batch normalization in deep learning. All four reviewers found significant merit in the ideas in the paper - but they also had a number of specific technical questions (e.g., by R1 and R4) that should be addressed in the final version of the paper.